# Metabolic, Affective and Neurocognitive Characterization of Metabolic Syndrome Patients with and without Food Addiction. Implications for Weight Progression

**DOI:** 10.3390/nu13082779

**Published:** 2021-08-13

**Authors:** Lucía Camacho-Barcia, Lucero Munguía, Ignacio Lucas, Rafael de la Torre, Jordi Salas-Salvadó, Xavier Pintó, Dolores Corella, Roser Granero, Susana Jiménez-Murcia, Inmaculada González-Monje, Virginia Esteve-Luque, Aida Cuenca-Royo, Carlos Gómez-Martínez, Indira Paz-Graniel, Laura Forcano, Fernando Fernández-Aranda

**Affiliations:** 1Department of Psychiatry, University Hospital of Bellvitge-IDIBELL, Hospitalet de Llobregat, 08907 Barcelona, Spain or lcamacho@idibell.cat (L.C.-B.); laarcreed_lm@hotmail.com (L.M.); or ilucas@idibell.cat (I.L.); sjimenez@bellvitgehospital.cat (S.J.-M.); 2CIBER Physiology of Obesity and Nutrition (CIBEROBN), Carlos III Health Institute, 28029 Madrid, Spain; RTorre@imim.es (R.d.l.T.); jordi.salas@urv.cat (J.S.-S.); xpinto@bellvitgehospital.cat (X.P.); dolores.corella@uv.es (D.C.); roser.granero@uab.cat (R.G.); inmaagonzalez@gmail.com (I.G.-M.); acuenca@imim.es (A.C.-R.); carlos.gomez@urv.cat (C.G.-M.); indiradelsocorro.paz@urv.cat (I.P.-G.); lforcano@imim.es (L.F.); 3Integrative Pharmacology and Neurosciences Systems, Institut Hospital del Mar d’Investigacions Mèdiques (IMIM), 08003 Barcelona, Spain; 4Department of Experimental and Health Sciences (CEXS-UPF), Universitat Pompeu Fabra, 08002 Barcelona, Spain; 5Universitat Rovira i Virgili, Department of Biochemistry and Biotechnology, Human Nutrition Unit, Reus, 43201 Tarragona, Spain; 6Institut d’Investigació Pere Virgili (IISPV), Reus, 43204 Tarragona, Spain; 7The Sant Joan University Hospital, Human Nutrition Unit, 43201 Reus, Spain; 8Lipids and Vascular Risk Unit, Internal Medicine, University Hospital of Bellvitge-IDIBELL, Hospitalet de Llobregat, 08907 Barcelona, Spain; vesteve@bellvitgehospital.cat; 9Department of Clinical Sciences, School of Medicine and Health Sciences, University of Barcelona, Hospitalet de Llobregat, 08907 Barcelona, Spain; 10Department of Preventive Medicine, University of Valencia, 46010 Valencia, Spain; 11Department of Psychobiology and Methodology, Autonomous University of Barcelona, 08193 Barcelona, Spain

**Keywords:** food addiction, metabolic syndrome, neurocognitive state, depression, quality of life

## Abstract

According to the food addiction (FA) model, the consumption of certain types of food could be potentially addictive and can lead to changes in intake regulation. We aimed to describe metabolic parameters, dietary characteristics, and affective and neurocognitive vulnerabilities of individuals with and without FA, and to explore its influences on weight loss progression. The sample included 448 adults (55–75 years) with overweight/obesity and metabolic syndrome from the PREDIMED-Plus cognition sub-study. Cognitive and psychopathological assessments, as well as dietary, biochemical, and metabolic measurements, were assessed at baseline. Weight progression was evaluated after a 3-year follow up. The presence of FA was associated with higher depressive symptomatology, neurocognitive decline, low quality of life, high body mass index (BMI), and high waist circumference, but not with metabolic comorbidities. No differences were observed in the dietary characteristics except for the saturated and monounsaturated fatty acids consumption. After three years, the presence of FA at baseline resulted in a significantly higher weight regain. FA is associated with worse psychological and neurocognitive state and higher weight regain in adults with metabolic syndrome. This condition could be an indicator of bad prognosis in the search for a successful weight loss process.

## 1. Introduction

According to the food addiction (FA) model, the consumption of certain types of food (sugary, salty, fatty, and processed) could be potentially addictive, presenting similarities with substance use disorders (SUD) by activating brain reward systems [1]. It has been suggested that the continuous intake of these high palatable foods can lead to changes in the food intake regulation [2], and could explain people’s difficulty to attach to other healthy dietary patterns [3] and overeating.

It has been found a higher prevalence of FA in individuals with obesity (18–24%) [4,5,6,7], than in normal-weight population (2–12%) [8,9,10,11,12].

Even though the prevalence of FA has been widely studied, its association with the obesity-related comorbidities remains unclear. In a sample of individuals eligible for obesity surgery with a body mass index (BMI) of ≥35 kg/m^2^, FA was not associated with metabolic comorbidities such as type 2 diabetes (T2D), hypertension, or non-alcoholic fatty liver disease [13]. Nonetheless, there is no available information about this association among an elder population with overweight and class 1 obesity. Evidence regarding the association between metabolic biochemical parameters and FA is also scarce. A previous study has shown that fasting plasma glucose levels were lower in patients with FA. However, the levels of serum insulin, homeostasis model assessment of insulin resistance (HOMA-IR), glycated hemoglobin (HbA1c), and the lipid serum profile were comparable in participants with and without FA [14].

FA may also affect food choice and calorie intake. It has been previously reported that higher FA scores were associated with higher energy intake [15], and that individuals with FA had higher dietary fat intake compared to those without FA [1] and higher BMI [16].

Besides the influence of FA in biological processes, it has been also related with other psychiatric disorders, such as eating disorders [4,17,18,19,20,21], anxiety [22], and behavioral addictions [23], being suggested as a transdiagnostic construct that underlines different conditions [24,25]. Particularly, higher depressive symptomatology has been found in the population with FA [18,22,24,26], presenting a positive association between both. Additionally, FA has been associated with impairments in quality of life, involving psychological, physical, and social aspects in both clinical [27] and non-clinical populations [28,29].

FA in individuals with obesity has also been related to impaired executive functions, such as decision making or attentional capacity [30] and cognitive impairment [31]. However, as far as we know, no study has measured the neurocognitive state related to FA in an advanced age population with metabolic syndrome.

The aims of the present study are threefold: (i) to describe metabolic parameters and dietary characteristics of individuals with and without FA, (ii) to describe affective and neurocognitive underlying vulnerabilities of individuals with and without FA, and (iii) to explore whether FA status, presence (FA+)/absence (FA−), influences weight loss progression. After reviewing the existing evidence, we hypothesize that those individuals with FA will present the worst metabolic state, more depressive symptomatology, and worse neurocognitive performance, as well as showing a worse weight loss progression in contrast with those without FA.

## 2. Materials and Methods

### 2.1. Study Design and Population

The PREDIMED-Plus (PREvención con DIeta MEDiterránea Plus) is a 6-year ongoing, multicenter, randomized parallel-group, intervention study with the primary objective of assessing the effects of an intensive lifestyle intervention with an energy-reduced Mediterranean diet (MedDiet), physical activity promotion, and behavioral support on primary prevention of cardiovascular disease (CVD) events. The PREDIMED-Plus study total cohort includes 6874 participants, recruited in 23 different Spanish centers, and randomly assigned to either an intensive weight loss intervention group, based on an energy-restricted MedDiet with physical activity promotion and behavioral support, or to a control group, advised to follow an ad libitum MedDiet without any other indication. The sample included men aged between 55 and 75 years and women between 60 and 75 years old, with overweight or obesity that met at least three components of metabolic syndrome at the moment of enrolment. The cohort description in detail has been previously portrayed [32] and the protocol is available in http://predimedplus.com/ (accessed on 13 August 2021). The trial was registered in 2014 at the International Standard Randomized Controlled Trial (ISRCT) with number ISRCT89898870. According to the ethical standards of the Declaration of Helsinki by the Research Ethics Committees, all the participating institutions approved the study protocol and procedures (PR240/13; H1509263926814; 2013/5276/I; 13-07-25/7proj2). All participants provided written informed consent. The current analysis was performed in a subsample of the original population, including *n* = 448 individuals from the PREDIMED-Plus cognition subprogram recruited in four different centers: Universitat Rovira i Virgili (Tarragona, Spain), Universidad de Valencia (Valencia, Spain), Institut Hospital del Mar d’Investigacions Mèdiques-IMIM (Barcelona, Spain), and Bellvitge University Hospital-IDIBELL (Barcelona, Spain).

### 2.2. Dietary Assessment

Dietary information was collected by trained dietitians at baseline and yearly thereafter during the follow-up visits. Total energy, macro, and micronutrients intake were assessed by a semi-quantitative 143-item food frequency questionnaire (FFQ) and estimated using Spanish food composition tables [33,34].

### 2.3. Biochemical, Anthropometric and Blood Pressure Measurements

Trained personnel assessed duplicated weight, height, and waist circumference at baseline, and subsequently at each of the follow-up visits. Body mass index (BMI, kg/m^2^) was later calculated using this information. Fasting blood samples were collected in order to perform biochemical analyses. Biochemical measurements included fasting plasma glucose, HbAc1 and insulin, albumin, transaminase enzymes, and lipid profile (total cholesterol, HDL-c, LDL-c and triglycerides). Blood pressure was measured 3 times using a validated semiautomatic oscillometer (Omron HEM-705CP). HOMA-IR was estimated with the formula: fasting glucose levels [mg/dL] × fasting insulin levels [µU/mL]/405,13 [35].

### 2.4. Psychometric Measures

Psychometric assessment was conducted by self-report questionnaires.

Yale Food Addiction Scale (YFAS) [3] has been validated in Spanish population [5]. It is a 25-item self-report questionnaire for measuring addictive eating behaviors, based on the Diagnostic and Statistical Manual of Mental Disorders (DSM-IV-TR) [36] criteria for substance dependence, adapted to the context of food consumption. Two scoring options are used to indicate the experience of addictive eating behavior: a dimensional (Likert scale) and a binary score. FA is diagnosed when at least three symptoms and a clinically significant impairment or distress are present within the previous 12 months. The internal consistency of the YFAS in our sample was α = 0.914.

The Beck Depression Inventory–II (BDI-II) [37] has been validated in Spanish general population [38]. It is a self-report questionnaire of 21 items that assess the severity of depressive symptoms in adults and adolescents, reflecting the diagnostic criteria for major depressive disorder listed in the DSM-5 [39]. Scores for each item range from 0 to 3, being the total score the sum of all responses. The standardized cut-offs are follows: 0–13 indicates minimal depression, 14–19 mild depression, 20–28 moderate depression, 29–63 severe depression. The internal consistency of the BDI-II in our sample was α = 0.873.

Short Form-36 Health Survey (SF-36) [40] has been validated in Spanish population [41]. It is a self-report questionnaire of 36 items in its short form. It is a generic measure of health status, through eight subscales: physical functioning, physical role, bodily pain, general health, vitality, social functioning, emotional role, and mental health. The higher the score, the better general quality of life. The internal consistency of the SF-36 in our sample was α = 0.933.

### 2.5. Cognitive Assessment

Montreal Cognitive Assessment (MoCA) [42] is a neurocognitive screening tool to detect mild cognitive impairment. It takes 7 to 10 min to be administered by healthcare professionals. The test assesses multiple cognitive functions, including short-term memory, visuospatial abilities, executive functioning, phonemic fluency, attention, concentration, working memory, language, and orientation. The test score ranges from 0 to 30, with higher scores indicating better performance. If the years of education are below 12, one point should be added to the final score. The test has been validated previously in Spanish population [43].

### 2.6. Statistical Analysis

Statistical analysis was carried with Stata17 for Windows [44]. For this analysis, we used the PREDIMED-Plus sub-cognition study database of January 14th, 2021.

Comparison between the groups were done with the chi-squared test (χ^2^) categorical variables and the T-test for independent groups for quantitative variables. Effect size was measured with the standardized Cohen’s d coefficient for mean differences and Cohen’s h for proportion differences (effect size was considered null for |d| < 0.20 or |h| < 0.20, low-poor |d| > 0.20 or |h| > 0.20, moderate–medium for |d| > 0.50 or |h| > 0.50, and large–high for |d| > 0.80 or |h| > 0.80) [45,46] In addition, Finner’s method was used to control the increase in the Type-I error due to the application of multiple null-hypothesis significance tests (this is a stepwise multiple test procedure aimed to adjust *p*-values controlling the familywise error rate, FWER) [47]. Missing data was handled by a multiple imputation (MI) procedure implemented in Stata17, which provides unbiased estimate parameters and their respective standard errors using Rubin’s combination rules and making no assumption about the missing-data mechanism [48,49].

The evolution of the BMI during the study was analyzed with 2×4 mixed analysis of variance (ANOVA), defining the group as the between-subjects factor (FA− versus FA+) and the time as the within-subjects factor (baseline, 1-yr, 2-yr, and 3-yr of the follow-up).

## 3. Results

### 3.1. Descriptive for the Sample

Table 1 displays the distribution for the total sample and the comparison for the patients with FA− and FA+. Among the total sample, the distribution of sex was 217 men (48.4%) versus 231 women (51.6%). Most participants were born in Europe (98.2%), were married, or lived with a stable partner (78.3%), achieved primary education levels (53.6%), and were retired (64.3%). Regarding the group of weight, the greatest likelihood was being in the obesity-I level (48.7%). Regarding comparison between the groups with the FA negative versus positive screening scores, differences only were found for the group of weight: FA+ associated with higher obesity levels. 

### 3.2. Comparison of Metabolic and Dietary Measures

Table 2 contains the results of the comparison between the groups for metabolic and dietary measures. The patients within the FA+ condition reported a higher mean for weight, BMI, and waist circumference. No differences were observed in the lipid profile, in the parameters of glucose metabolism, liver function, blood pressure values. and physical activity total energy expenditure. Regarding dietary information, participants with FA had a higher intake of monounsaturated fatty acids (MUFAs) (grams and percentage estimates), saturated fatty acids (SFAs) (grams and percentage estimates), and grams per day of trans fatty acids.

### 3.3. Comparison of Psychological and Neuropsychological Measures

Table 3 contains the results of the comparison between the groups for the BDI, SF-36, and the MoCa scores. The presence of FA+ was associated with worse psychological state, except for the SF-36 physical pain domain.

### 3.4. Evolution of the BMI during the Study

Figure 1 shows the line graph with the means registered for the BMI at the beginning of the study and during the three years of the follow-up. Appendix A contains the complete results of the 2×3 ANOVA. A significant interaction group-by-time was found (F (df = 2.431) = 4.18, *p* = 0.021, η2 = 0.010), suggesting that the progression of the BMI during the study was different in each FA condition. Concretely, polynomial contrasts evidenced a quadratic trend within the FA− condition (*p* = 0.004, η2 = 0.020), while no linear-quartic trend emerged within the FA+ condition. Contrasts for the factor group (FA+ versus FA− comparisons) also achieved significant results, with higher mean scores among patients with FA+ at baseline (*p* = 0.001, η2 = 0.034) and at three years of the follow-up (*p* = 0.001, η2 = 0.033).

## 4. Discussion

The present study aimed to describe metabolic parameters and dietary characteristics, as well as affective and neurocognitive underlying vulnerabilities, of individuals with and without food addiction, and to explore whether FA status (presence/absence) influences weight loss progression, in a metabolic syndrome at high cardiovascular risk population.

The prevalence of FA in our sample reached 5.8%, underneath the number of the general population that has been reported to be about 8% [50]. However, information of the prevalence of FA among the elderly population is scarce. Data from the Nurses’ Health Studies (NHS) cohort, a sample of women aged between 60 and 88 years old, showed that the number of cases of FA reached 2.7%. This prevalence was strongly linked with age, reaching 6% in the age range of 60–65 years, and continually decreased till 1.4% in ages between 75 and 88 [51]. The decline in the prevalence of FA diagnosis while increasing in age could be associated with several physiological alterations related to the ageing process, including changes in the taste and smell senses [52] that could be affecting the motivation for palatable food, as well as changes in the physiology of several endocrinal factors such as ghrelin, cholecystokinin, and leptin that may result in a relevant decrease in appetite and food intake [53].

As our first aim, we explored the metabolic status and the presence of comorbidities. In our study, the diagnosis of FA was associated with higher total weight, BMI, and higher rates of morbid obesity (BMI ≥ 35 kg/m^2^). Previous findings have reported similar associations [51,54,55]. Yet, in our study, no link was observed between the presence of FA and obesity-related comorbidities. This lack of association may be due to the general characteristics of the study population, individuals with overweight or obesity that met at least three components of metabolic syndrome at the moment of enrolment, and therefore at high cardiovascular risk. Nonetheless, we did observe a significant difference in the waist circumference values, an indicator of abdominal obesity and a predictor of metabolic disorders such as T2D and CVD [56].

It is worth mentioning that, even though the presence of FA could be considered a contributor factor in the weight gain and latter development of obesity, the fat mass accumulation and the obesity-related comorbidities are affected by several other influences, such as the dietary pattern, the metabolic state, genetics, and epigenetics, among others [57].

Regarding the dietary characteristics of our sample, we were not able to identify any differences in-between the macronutrient dietary profile and the total calorie intake of the FA and non-FA participant groups. Nevertheless, we did observe significant differences in the fatty acid distribution, specifically in the saturated and monounsaturated fatty acid consumption, as well as in the trans fatty acid intake. The same differences were described in previous findings, where individuals with obesity and FA exhibited a significantly higher consumption of fat subcomponents: saturated, monosaturated, polyunsaturated, and trans fats [58]. Food with a high content of fat, especially saturated and trans fatty acids, are highly palatable and more likely to be excessively consumed and to be associated with compulsive overeating [59].

In relation to our second aim, higher depressive symptomatology was found in those patients presenting FA. The presence of depression among metabolic syndrome patients have already been mentioned in the literature [60,61,62,63], as well as the association between FA and depression [18,22,24,26]. A possible common feature between both comorbidities could be the use of the eating behavior as a way to regulate negative emotional states that may lead to overeating [64], as well as to changes in the food intake regulation [2], establishing a continuous feedback between each other [10]. Depression has been associated with higher daily caloric intake [64], and high adipose mass has been associated with several metabolic disturbances that are implicated in the control of emotions and mood [65,66]. This may explain the increase of depression in our sample, from minimal depression in those without FA, to mild depression in those with FA, according to the standardized cut-offs [37].

In a similar way, according to what it was hypothesized, those patients with FA presented higher impairments in all the assessed parameters of quality of life than those without FA, except for the one related to physical pain. Impairments in quality of life have been already mentioned in the literature in populations that present FA [27,28,29], but just a few studies have explored the influence of FA over the quality of life in contrast with the other metabolic parameters. Its findings are in concordance with the ones of the present study, being that the metabolic disturbances seem not to be enough to imply detriments in quality of life [14], but the presence of FA does. As has been mentioned in the literature, addictive processes, both behavior- and substance use-related, decreases quality of life [67].

Regarding the neurocognitive assessment, our results showed that FA+ group presented lower scores in a general cognitive state test than the group without FA, pointing out that FA was related to a more pronounced cognitive decline in an older adult population with metabolic syndrome. Fast food intake in elderly population has been already associated with higher incidence of mild cognitive impairment [68], as well as the preference for processed foods with poorer cognitive performance and impaired executive functioning in this population [69]. In contrast, higher adherence to the Mediterranean diet has been associated with a reduced risk for developing mild cognitive impairment in nondemented elderly participants [70]. Our results may suggest that the presence of FA patterns could be associated with an increased risk of cognitive decline within this population.

Finally, one of the most relevant contributions of this study is the longitudinal assessment of the effect of FA on the evolution of weight at three years. Our results indicate that the presence of FA at baseline resulted in a significant higher weight regain after three years, when following a Mediterranean diet. These findings are consistent with prior research that observed an association between the presence of FA and worse results while seeking a weight loss [71,72,73]. Previous results from our group showed that among individuals with obesity seeking bariatric surgery, the presence of FA at baseline predicted poorer weight loss achievement after a dietary and lifestyle intervention before the surgical intervention [6].

These findings may suggest that the presence of FA could be an indicator of bad prognosis on the search for a successful weight loss process, influencing the final outcome in the treatments. Nonetheless, there is also evidence that the FA symptoms can diminish after a weight reduction [74,75]. Therefore, taking all this into account, even though the presence of FA could be considered a disadvantage at the moment of reaching a successful result, implementing additional and specialized support to these individuals during the treatment may be a good strategy for obtaining better results [25].

### Limits and Strengths

Our study has some limitations that should be considered. The study cohort, a Mediterranean older adult population with metabolic syndrome at high cardiovascular risk, is not representative of the general population, and therefore our results cannot be extrapolated. As in any observational study, the possibility of residual or unmeasured confounding could not be eliminated. Additionally, even though FFQs are suitable tools for epidemiological studies’ dietary assessment, being a subjective method, we cannot disregard some errors in the estimation of the dietary information.

Despite its limitations, this study adds a characterization of a population that has not been previously thoroughly described, a senior population with food addiction symptomatology.

## 5. Conclusions

As general a conclusion, the results of this study indicated that the presence of FA is associated with higher depressive symptomatology, worse neurocognitive state, and lower quality of life than with metabolic parameters. Additionally, those participants with FA had higher weight regain than those without FA after three years of follow-up. All these factors may be indicators of worse prognosis for metabolic syndrome patients that also present FA, for which the present results may be helpful in order to establish better treatment approaches. It is possible to suggest that the population with FA could benefit from both a diet-based treatment as well as a specific one targeting the addictive process, which could improve the psychological, social, and neurological functioning of the patients.

## Figures and Tables

**Figure 1 nutrients-13-02779-f001:**
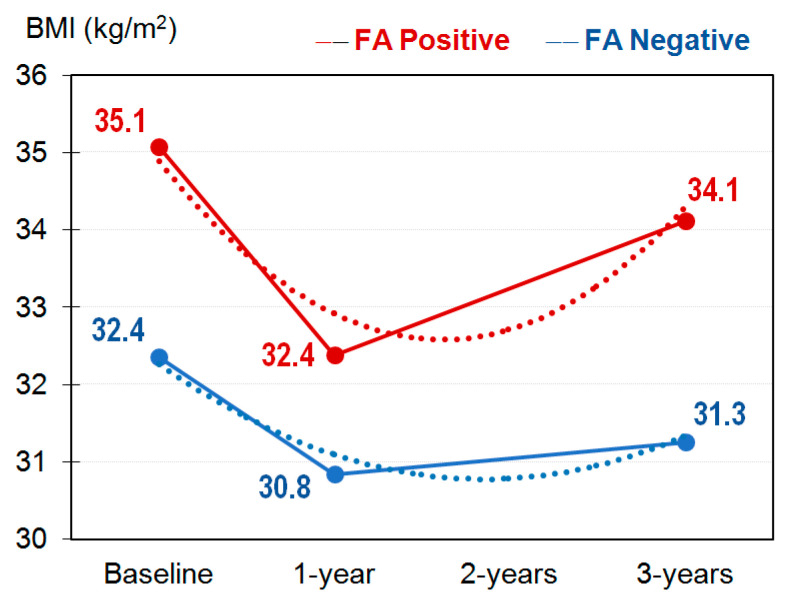
Evolution of the BMI (kg/m^2^) during the study. Note. Dash line: polynomial quadratic trend. Sample size: *n* = 434.

**Table 1 nutrients-13-02779-t001:** Descriptive for the sample at baseline.

		Total Sample (*n* = 448)		FA Negative (*n* = 422)		FA Positive (*n* = 26)				
		*n*	%	*n*	%	*n*	%	χ^2^	df	*p*
Sex	Male	217	48.4%	208	49.3%	9	34.6%	2.11	1	0.146
	Female	231	51.6%	214	50.7%	17	65.4%			
Origin	Europe	440	98.2%	415	98.3%	25	96.2%	0.67	1	0.414
	South America	8	1.8%	7	1.7%	1	3.8%			
Civil status	Single	17	3.8%	16	3.8%	1	3.8%	4.14	3	0.247
	Married	351	78.3%	333	78.9%	18	69.2%			
	Divorced-separated	30	6.7%	29	6.9%	1	3.8%			
	Widowed	50	11.2%	44	10.4%	6	23.1%			
School	University (high)	42	9.4%	38	9.0%	4	15.4%	1.39	3	0.708
	University (grade)	37	8.3%	35	8.3%	2	7.7%			
	Secondary	129	28.8%	123	29.1%	6	23.1%			
	Primary	240	53.6%	226	53.6%	14	53.8%			
Employment	Unemployed	80	17.9%	76	18.0%	4	15.4%	1.48	4	0.831
	Work at home	46	10.3%	43	10.2%	3	11.5%			
	Retired	288	64.3%	271	64.2%	17	65.4%			
	Unemployed (incomes)	21	4.7%	19	4.5%	2	7.7%			
	Unemployed (no-incomes)	13	2.9%	13	3.1%	0	0.0%			
Group weight	Over-weight	123	27.5%	122	28.9%	1	3.8%	14.64	3	**0.002 ***
	Obesity I (BMI 30-35)	218	48.7%	206	48.8%	12	46.2%			
	Obesity II (BMI 35–40)	103	23.0%	90	21.3%	13	50.0%			
	Obesity III (BMI >40)	4	0.9%	4	0.9%	0	0.0%			
		Mean	SD	Mean	SD	Mean	SD	F	df	*p*
Age, years-old		65.25	4.63	65.22	4.63	65.73	4.64	0.30	1.446	0.582

Note. SD: standard deviation. df: degrees of freedom. * Bold: significant comparison.

**Table 2 nutrients-13-02779-t002:** Comparison of metabolic and dietary measures.

	FA Negative (*n* = 422)		FA Positive (*n* = 26)			
	Mean	SD	Mean	SD	*p*	|d|
Total cholesterol, mg/dL	207.70	40.10	215.00	36.63	0.366	0.19
Triglycerides, mg/dL	160.81	77.69	155.08	81.02	0.716	0.07
LDL cholesterol, mg/dL	125.65	33.69	128.58	32.47	0.666	0.09
HDL cholesterol, mg/dL	51.11	12.95	55.46	10.02	0.093	0.38
Albumin, g/dL	4.43	0.50	4.52	0.24	0.401	0.21
Glucose, mg/dL	116.81	30.18	110.81	18.53	0.317	0.24
Insulin, mIU/ml	18.68	8.64	18.36	8.37	0.855	0.04
HOMA-IR	5.45	3.08	5.13	2.66	0.607	0.11
HbA1c, %	6.13	0.76	6.10	0.81	0.849	0.04
Alanine aminotransferase, U/L	26.95	12.60	23.20	11.10	0.139	0.32
Aspartate aminotransferase, U/L	24.12	8.45	22.85	6.81	0.454	0.17
Systolic blood pressure, mm Hg	140.75	15.03	139.69	13.21	0.725	0.07
Diastolic blood pressure, mmHg	79.93	9.50	79.73	9.76	0.918	0.02
Physical activity total energy expenditure, MET·min/week	849.05	801.20	769.23	832.43	0.623	0.10
Weight, kg	85.42	13.39	91.29	14.50	**0.031 ***	0.42
BMI, kg/m^2^	32.38	3.39	35.06	3.07	**<0.001 ***	**0.83 ^†^**
Waist circumference, cm	107.53	10.15	111.88	11.02	**0.035 ***	0.41
Total energy intake, kcal/day	2398.66	573.70	2527.12	815.43	0.282	0.18
Carbohydrate, g/d	241.40	78.35	245.12	92.33	0.816	0.04
Carbohydrate, %	39.93	6.53	38.48	7.04	0.276	0.21
Protein, g/d	100.57	20.83	104.97	26.46	0.304	0.19
Protein, %	17.10	2.90	17.20	2.87	0.869	0.03
Total fat, g/d	107.66	28.71	119.47	46.04	0.052	0.31
Total fat, %	40.52	6.12	42.55	5.66	0.099	0.35
SFAs, g	27.69	9.21	31.65	12.89	**0.039 ***	0.35
SFAs, %	10.33	1.92	11.15	1.99	**0.034 ***	0.42
MUFAs, g/d	55.28	14.98	63.53	25.87	**0.010 ***	0.39
MUFAs, %	20.91	4.20	22.70	3.75	**0.035 ***	0.45
PUFAs, g/d	18.12	6.58	19.58	8.19	0.280	0.20
PUFAs, %	6.80	1.80	6.93	1.41	0.710	0.08
Trans fatty acids, g/d	0.65	0.41	0.87	0.50	**0.009 ***	0.48
Prevalence	n	%	n	%	*p*	|h|
Hypertension	317	75.1%	23	88.5%	0.123	0.35
Diabetes	121	28.7%	6	23.1%	0.539	0.13
Hypercholesterolemia	216	51.2%	13	50.0%	0.907	0.02

Note. SD: standard deviation, MET, metabolic equivalent of task, SFA: saturated fatty acids, MUFA: monounsaturated fatty acids, PUFA: polyunsaturated fatty acids. * Bold: significant comparison. ^†^ Bold: effect size into the range mild–moderate to high–large.

**Table 3 nutrients-13-02779-t003:** Comparison of psychological measures.

	FA Negative (*n* = 422)		FA Positive (*n* = 26)			
	Mean	SD	Mean	SD	*p*	|d|
BDI total score	7.97	6.42	14.92	9.38	**<0.001 ***	**0.87 ^†^**
SF-36 total score	78.34	16.35	63.46	18.33	**<0.001 ***	**0.86 ^†^**
SF-36 physical function	75.84	18.98	63.27	18.05	**0.001 ***	**0.68 ^†^**
SF-36 physical role	77.86	33.33	48.08	41.79	**<0.001 ***	**0.79 ^†^**
SF-36 physical pain	70.43	22.54	62.31	21.41	0.075	0.37
SF-36 general health	76.21	16.78	66.92	17.27	**0.007 ***	**0.55 ^†^**
SF-36 vitality	62.73	20.28	46.15	22.06	**<0.001 ***	**0.78 ^†^**
SF-36 social function	92.48	14.29	84.73	19.11	**0.009 ***	0.46
SF-36 emotional role	87.92	28.19	67.96	42.69	**0.001 ***	**0.55 ^†^**
SF-36 mental health	75.10	18.83	64.31	17.89	**0.005 ***	**0.59 ^†^**
MoCA total	23.58	4.31	21.42	6.54	**0.017 ***	0.39

* Bold: significant comparison. ^†^ Bold: effect size into the range mild–moderate to high–large.

## Data Availability

Due to signed consent agreements regarding data sharing, there are restrictions on data availability for the PREDIMED-Plus trial. These only allow access to external researchers for studies following the project purposes. Requestors wishing to access the PREDIMED-Plus trial data used in this study can make a request to the PREDIMED-Plus trial Steering Committee chair: jordi.salas@urv.cat. The request will then be passed to members of the PREDIMED-Plus Steering Committee for deliberation.

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
