# Peer review of "Metabolic, Affective and Neurocognitive Characterization of Metabolic Syndrome Patients with and without Food Addiction. Implications for Weight Progression"

_nutrients, 2021, doi:10.3390/nu13082779_

Round 1
Reviewer 1 Report
The article written by Lucía Camacho-Barcia et al. describes metabolic, affective and neurocognitive characterization of metabolic syndrome patients with and without Food Addiction.
minor points
1. line 59, 243 - BMI is given without units. This unit is written in Table 2. Add it also to introduction and discussion. Moreover, please note it correctly, i.e. kg/m2 not kg/m2 (Table 2, Figure 1).
2. HbA1c and HOMA-IR abrreviations are explained in lines 124 and 127 respectively. However, they are used earlier in the introduction (line 65) without explanation. Please correct it.
3. The conclusion section should be a little developed.
Author Response
Please see the attachment
Dear Reviewer
Thank you for your comments. As suggested, we have attached a revised version of the manuscript entitled “Metabolic, affective and neurocognitive characterization of metabolic syndrome patients with and without Food Addiction. Implications for weight progression" (Manuscript ID: nutrients-1319424).
We have made changes to the manuscript according to your comments, using the “Track Changes” function, and relevant changes have been written as answer´s to the comments as well in the present document. The manuscript has been prepared according to the journal's instructions.
Reviewer #1:
Comment 1:
- line 59, 243 - BMI is given without units. This unit is written in Table 2. Add it also to introduction and discussion. Moreover, please note it correctly, i.e. kg/m2 not kg/m2 (Table 2, Figure 1).
Response: Thank you very much for your observation, this has been corrected.
Comment 2:
- HbA1c and HOMA-IR abrreviations are explained in lines 124 and 127 respectively. However, they are used earlier in the introduction (line 65) without explanation. Please correct it.
Response: Thank you very much for your observation, this has been corrected.
Comment 3:
- The conclusion section should be a little developed.
Response: Thank you very much for your comment, this section has been updated as follows:
As general a conclusion, the results of this study indicated that the presence of FA is associated with higher depressive symptomatology, worse neurocognitive state and lower quality of life, than with metabolic parameters. Additionally, those participants with FA had higher weight regain than those without FA after three years of follow-up. All these factors may be indicators of worse prognosis for metabolic syndrome patients that also present FA, for which the present results may be helpful in order to establish better treatment approaches. It is possible to suggest that the population with FA could benefit from both, a diet-based treatment, as well as a specific one target the addictive process, which could improve the psychological, social and neurological functioning of the patients.

Reviewer 2 Report
This paper assessed the role of metabolic, dietary, cognitive and psychological factors associated with food addiction (FA) via baseline data in a cohort assigned to a lifestyle intervention. Weight loss in FA groups was compared to those without FA.
Overall, this paper is well-written and clearly defined in its objectives and outcomes. However, the paper’s impact may be improved by further analysis of BMI in FA and non-FA subjects with respect to the above mentioned factors.
Detailed Comments:
Results:
Table 2: It may also be revealing to assess total intake of sugar/d (which may be more indicative of appetizing foods preferred by FA groups)
Table 3: how much of Table 3 psychological and cognitive associations is influenced by obesiy/overweight prevalence in FA? Were there analyses examining the role of BMI with respect to the measures from Table 3?
Discussion:
The findings that FA subjects intake more SFA,TFA and MUFA but show no associations to metabolic dysfunction biomarkers would benefit from more discussion. Is this because the increased intake is not clinically meaningful eg increasing risk of elevated biomarkers?
The finding that kcal intake/d is not different in FA and non-FA groups while BMI is significantly different suggests an issue with intake self-report. Please comment on this further. At the very least add this as a limitation.
Page 8 line 274, “Which may explain the increase of depression in our sample, from minimal de-274 pression in those without FA, to mild depression in those with FA, according to the standardized cut-offs” sentence is not grammatically correct, please revise ‘which’ to ‘this’.
Page 8, line 278, ‘excepted’ is not used correctly, please revise
Page 8, line 279, ‘affectations’ is not used correctly, please revise
Page 8, line 284 “As it has been mentioned in the literature, 284 addictive process, behavioral and substance use related, decreases quality of life” this sentence is not grammatically correct, please revise
